

# Feeding intensity of insect herbivores is associated more closely with key metabolite profiles than phylogenetic relatedness of their potential hosts

Carole B. Rapo[1,2,3], Urs Schaffner[2], Sanford D. Eigenbrode[3], Hariet L. Hinz[2], William J. Price[4], Matthew Morra[5], John Gaskin[6] and Mark Schwarzländer[3]

[1] Climate-KIC Office, Swiss Federal Institute of Technology, Zürich, Switzerland
[2] CABI Switzerland, Delemont, Switzerland
[3] Department of Entomology, Plant Pathology and Nematology, University of Idaho, Moscow, ID, USA
[4] Statistical Programs, University of Idaho, Moscow, ID, USA
[5] Soil and Water Systems, University of Idaho, Moscow, ID, USA
[6] Northern Plains Agricultural Research Laboratory, USDA ARS, Sidney, MT, USA

Corresponding author
Sanford D. Eigenbrode,
sanforde@uidaho.edu

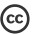

## ABSTRACT

Determinants of the host ranges of insect herbivores are important from an evolutionary perspective and also have implications for applications such as biological control. Although insect herbivore host ranges typically are phylogenetically constrained, herbivore preference and performance ultimately are determined by plant traits, including plant secondary metabolites. Where such traits are phylogenetically labile, insect hervivore host ranges are expected to be phylogenetically disjunct, reflecting phenotypic similarities rather than genetic relatedness among potential hosts. We tested this hypothesis in the laboratory with a Brassicaceae-specialized weevil, *Ceutorhynchus cardariae* Korotyaev (Coleoptera: Curculionidae), on 13 test plant species differing in their suitability as hosts for the weevil. We compared the associations between feeding by *C. cardariae* and either phenotypic similarity (secondary chemistry—glucosinolate profile) or genetic similarity (sequence of the chloroplast gene *ndh*F) using two methods—simple correlations or strengths of association between feeding by each species, and dendrograms based on either glucosinolates or *ndh*F sequence (i.e., a phylogram). For comparison, we performed a similar test with the oligophagous *Plutella xylostella* (L.) (Lepidoptera: Plutellidae) using the same plant species. We found using either method that phenotypic similarity was more strongly associated with feeding intensity by *C. cardariae* than genetic similarity. In contrast, neither genetic nor phenotypic similarity was significantly associated with feeding intensity on the test species by *P. xylostella*. The result indicates that phenotypic traits can be more reliable indicators of the feeding preference of a specialist than phylogenetic relatedness of its potential hosts. This has implications for the evolution and maintenance of host ranges and host specialization in phytophagous insects. It also has implications for identifying plant species at risk of nontarget attack by potential weed biological control agents and hence the approach to prerelease testing.

# INTRODUCTION

Phylogenetic constraints are evident in the host range of many specialized organisms, including plant pathogens (*Gilbert & Webb, 2007*), parasitic Hymenoptera (*Desneux et al., 2012*), and insect herbivores (*Bernays & Chapman, 1994*). These patterns likely reflect the morphological and biochemical similarities among closely related potential host species (*Krasnov et al., 2004*; *Poulin, 2005*), accounting for phylogenetic associations between lineages of insect herbivores and plant families with distinct chemical defenses (*Ehrlich & Raven, 1964*), and host ranges of individual insect species among their potential hosts. For example, *Rasmann & Agrawal (2011)* reported that total cardenolide concentration among 18 *Asclepias* spp. was associated both with their relatedness to *Asclepias syriaca* and with survival of larvae of *Tetraopes tetraophthalmus* (Coleoptera: Chrysomelidae), a specialist on *A. syriaca*, implicating total cardenolide concentration as a factor accounting for the constrained host range of *T. tetraophthalmus*. However, apart from *Rasmann & Agrawal (2011)*, phytochemical phenotypic similarity consistent with the phylogenetically constrained host range of an insect herbivore seems rarely to have been examined.

On the other hand, some insect herbivores have phylogenetically disjunct host ranges in which plants closely related to a primary host are not attacked while more distantly related ones are (*Hinz, Schwarzländer & Gaskin, 2008*; *Weiblen et al., 2006*). Such disjunct host ranges may also be phytochemically determined when phytochemistry diverges from phylogenetic relatedness. For example, although major classes of plant secondary compounds are associated with particular plant families, there are exceptions (*Wink, 2003*) and these exceptions can explain divergent specialization, across plant families, and among confamilial species by a single herbivore (*Wahlberg, 2001*). Similarly, wide taxonomic host shifts by specialized insect herbivores are evidently driven by phytochemical or other phenotypic similarities among potential hosts rather than phylogenetic relatedness of the host plant species (*Kergoat et al., 2005*; *Murphy & Feeny, 2006*). At a finer taxonomic scale, phytochemical profiles can exhibit a weak phylogenetic signal within a compound class, varying among closely-related plant species (*Asres, Sporer & Wink, 2004*; *Becerra, Noge & Venable, 2009*; *Cacho, Kliebenstein & Strauss, 2015*; *Fahey et al., 2018*; *Olsen et al., 2016*; *Pearse & Hipp, 2009*). Individual compounds within a class can have divergent effects on herbivores (*Agrawal et al., 2011*; *Hopkins, Van Dam & Van Loon, 2009*; *Liu, Klinkhamer & Vrieling, 2017*), potentially accounting for disjunct host ranges of individual herbivore species.

In this article, we test whether feeding by the weevil *Ceutorhynchus cardariae* Korotyaev (Coleoptera: Curculionidae) on potential host plant species is better explained by the genetic similarity of these potential hosts to *Lepidium draba* L. (Brassicaceae), the weevil's major European field host, or by the similarity of the glucosinolate (hereafter GS) profiles of these hosts to that of *L. draba*. Previous work has shown that the host range of

*C. cardariae* is phylogenetically disjunct within Brassicaceae (*Hinz, Schwarzländer & Gaskin, 2008*). GS are secondary metabolites primarily occurring in Brassicaceae and serving both as defenses against generalists and as cues for host selection and acceptance by specialists (*Hopkins, Van Dam & Van Loon, 2009*; *Schoonhoven, Van Loon & Dicke, 2005*). GS have variable phylogenetic signal among species within a single genus (*Streptanthus*) (*Cacho, Kliebenstein & Strauss, 2015*) and within a tribe (Cardamineae) (*Olsen et al., 2016*) within Brassicaceae.

In addition to improving understanding of the mechanisms shaping host ranges of phytophagous insects, this study has practical implications. Detection of phylogenetically disjunct host ranges and understanding how they can occur is important for improving the environmental safety and effectiveness of weed biological control. The disjunct host range of *C. cardariae* was detected as part of prerelease testing of this species as a biological control candidate for *L. draba*, which is invasive in North America, Australia and elsewhere (CABI, https://www.cabi.org/isc/datasheet/10621). Prerelease testing of potential weed biological control candidates includes assessment of the potential for nontarget attack. Typically, plant species are selected for testing based on their relatedness to the target species under the assumption that host ranges are phylogenetically constrained (*Wapshere, 1974*), an approach that generally proves reliable (*Wheeler & Madeira, 2017*). However, phylogenetically disjunct host ranges (though rare) occur, so selection of plants for prerelease testing for biological control should be guided by additional criteria, including secondary metabolite profiles similar to the target weed, regardless of relatedness (*Schaffner, 2001*; *Wapshere, 1974*; *Wheeler, 2005*; *Wheeler & Schaffner, 2013*). This study examines that premise.

Finally, for comparison with *C. cardariae*, we assessed the relative strength of the associations between genetic similarity and the phenotypic (GS profile) similarity with feeding by the oligophagous herbivore, *Plutella xylostella* (L.) (Lepidoptera: Plutellidae). We hypothesized that, given its documented wider host range, *P. xylostella* feeding is less constrained than *C. cardariae* by GS profiles of its potential hosts.

## METHODS

### Insects

*Ceutorhynchus cardariae* were from a colony at CABI Switzerland, established from stock originally collected from Romania (H. Hinz, 2006, unpublished data) and is reared continuously on *L. draba*. The primary host of *C. cardariae* in its native range, which spans diverse climatic zones across Eurasia, is *L. draba;* it has also been found to attack the closely related European *L. campestre* L. (*Hinz & Diaconu, 2015*). From early to late spring, females feed and oviposit into petioles, leaf midribs and stems of *L. draba*. The larvae have three instars and elicit the formation of galls. When mature, larvae exit the galls to pupate in the soil (*Hinz & Diaconu, 2015*).

Eggs of *P. xylostella* were obtained from a commercial colony (Syngenta Corporation, Basel, Switzerland) and the larvae were reared in a growth chamber ($18 \pm 2\,°C$) on an artificial diet modified from *Biever & Boldt (1971)* (agar, casein, wheat germ, sodium propionate, sorbic acid, methyl paraben, vitamin mix (USDA), salt mix (Wesson), chlortetracycline).

**Table 1 Plant species used in this study.** Plant species used in this study, including nativity (EU = native to Europe, NA = native to North America), seed origin, year of collection, and life history (A = Annual, B = Biennial, P = Perennial). GenBank accession numbers are included after species name for DNA sequences used in this study.

| Plant species* | Nativity | Seed source** | Year | Life history |
|---|---|---|---|---|
| *Barbarea orthoceras* (MK982847) | NA | B & T World Seeds*, USA | 2007 | B, P |
| *Brassica nigra* (MK982848) | EU | Karlsruhe, Germany | 2007 | A |
| *Brassica nigra* (2nd pop) | EU | Nantes, France | 2007 | A |
| *Camelina microcarpa* (DQ288746.1) | EU | Wyoming, USA | 2004 | A, B |
| *Camelina microcarpa* (2nd pop) | EU | Wyoming, USA | 2004 | A, B |
| *Caulanthus inflatus* (MK982849) | NA | San Luis Obispo, California, USA | 2003 | A |
| *Draba nemorosa* (MK982850) | NA | Wyoming, USA | 2005 | A |
| *Hesperis matronalis* (DQ288776.1) | EU | Wyoming, USA | 2004 | B, P |
| *Lepidium campestre* (MK982851) | EU | Göttingen, Germany | 2007 | B |
| *Lepidium campestre* (2nd pop) | EU | Paris, France | 2007 | B |
| *Lepidium crenatum* (MK982852) | NA | Colorado, USA | 2006 | P |
| *Lepidium draba* (DQ288790.1) | EU | Idaho, Ada Co., USA | 2002 | P |
| *Lepidium draba* (2nd pop) | EU | Iasi, Romania | 2003 | P |
| *Lepidium latifolium* (MK982853) | EU | Tyana, Turkey | 2006 | P |
| *Lepidium latifolium* (2nd pop) | EU | California, USA | 2002 | P |
| *Lepidium squamatum* (MK982854) | EU | Karlsruhe, Germany | 2005 | A |
| *Lepidium squamatum* (2nd pop) | EU | Fribourg, Switzerland | 2006 | A |
| *Stanleya pinnata* (DQ288832.1) | NA | Flagstaff Native Plant and Seed*, USA | 2003 | P |
| *Stanleya viridiflora* (MK982855) | NA | Wyoming 14205, USA | 2005 | P |

Notes:
* 2nd populations were not used in bioassays, only for GS analysis.
** Seeds were from commercial sources; all those unmarked were collected from wild populations.

Artificial diet was used to avoid potential confounding effects of larval conditioning during feeding on a particular host plant species prior to bioassay. Third-instar *P. xylostella* were used for bioassays, based on pilot tests that indicated this stage was optimal for distinguishing differences in feeding among the test plant species. Under natural conditions, *P. xylostella* is oligophagous within Brassicaceae, with hosts that include *L. draba* (*Cripps et al., 2006*), the cultivated *Brassica* spp. (*Shelton & Nault, 2004*; *Talekar & Shelton, 1993*), and at least 40 wild species in the genera *Barbarea*, *Brassica*, *Lepidium*, *Hesperis*, *Erysimum*, *Capsella*, *Sinapis* and others (*Sarfraz et al., 2011*; *Talekar & Shelton, 1993*).

## Plants

One or two populations each from thirteen plant species, including *Lepidium draba*, were used in this study (Table 1). These species differ in their relatedness to *L. draba* and in their suitability as hosts for *C. cardariae*, based on no-choice feeding and development bioassays (*Hinz & Diaconu, 2015*; H. Hinz, 2006, unpublished data). Hosts (species on which at least some *C. cardariae* development occurred in no-choice tests) closely related to *L. draba* included *Barbarea orthoceras* Ledeb., *Lepidium campestre* (L.) W.T. Aiton, *Lepidium latifolium* (L.), and *Lepidium squamatum* Forssk. Hosts distantly related to *L. draba* included *Caulanthus inflatus* S. Watson and *Stanleya pinnata* (Pursh) Briton.

Non-hosts closely related to *L. draba* included *Camelina microcarpa* Andrz. ex DC. and *Lepidium crenatum* (Greene) Rydb. Non-hosts distantly related to *L. draba* included *Brassica nigra* (L.), *Draba nemorosa* (L.), *Hesperis matronalis* (L.), and *Stanleya viridiflora* Nutt. Plants were grown from seed in individual pots and maintained in a greenhouse, with 20–25 °C day and 10 °C night. Each population of the 13 plant species was represented by 10 mature individuals with distinguishable young and old leaves. A total of 6–8 days prior to bioassays or collecting tissue for chemical analysis, the plants were placed outside in a large screen cage to expose them to ambient light that could influence secondary metabolites (*Escobar-Bravo, Klinkhamer & Leiss, 2017*). Five individuals of one population each from the 13 plant species were then used for feeding bioassays and the other five for GS analysis. GS profiles were characterized on additional plant populations for six of the species (Table 1) to assess intraspecific GS variation within these plant species; five plants of each, grown at the same time and with the same protocol as those used for bioassays and were used for this analysis.

## Experimental design

The experiment was run as a randomized complete block design with five temporal blocks with one replicate of each plant per block (Table S1). Blocks (designated A–E) were separated by approximately 3 days. Block E was tested in two stages because not enough adults of *C. cardariae* were available at once, but these followed one another closely and were treated as a single complete block for analysis.

## Feeding bioassays

Feeding bioassays with *C. cardariae* and *P. xylostella* were performed between 7 April and 15 May 2008. For *C. cardariae*, only recently overwintered females were used, consistent with their typical life history. Feeding by females precedes oviposition and eventual host use. Prior to testing, *C. cardariae* females were placed individually into Petri dishes and starved for 24 h after which they were randomly assigned to treatments. For *P. xylostella*, larvae were used because they are the feeding stage. Second instars were taken directly from diet and not starved prior to testing because the diet is presumed not to introduce any preconditioning. Individuals of each insect species were placed into separate clip cages (1.5 × 2.8 cm) attached to adjacent or nearly adjacent fully expanded leaves in the upper portion of the same plant ($n = 5$ for each plant species), to offer leaves of the same developmental age to both insects at the same time. A total of 65 *C. cardariae* and 60 *P. xylostella* were used in the bioassays. The clip cages were positioned in the middle of the leaf with the cage over the adaxial surface. Plants with insects were kept at 19 °C in the laboratory under natural light regime during the bioassay. After 48 h, the number of feeding punctures produced by *C. cardariae* was counted and the leaves on which *P. xylostella* individuals fed were collected and preserved in a herbarium for later quantification of herbivory based on leaf area removed.

The area removed by *P. xylostella* was quantified based on scanned images of leaves used in bioassays. First, the preserved leaf material was placed in a hydrator for 24 h to relax the leaf tissue. Then, each individual leaf was imaged with a digital scanner (Hp Scanjet

8290). Using the software GIMP 2.6., we selected the part of the leaf on the digital image that was exposed in the clip cage to *P. xylostella*. That portion of the image was then loaded into the Compu Eye Leaf and Symptom Area Software (http://www.ehabsoft.com/CompuEye/LeafSArea/ (*Bakr, 2005*)) to measure the feeding damage in mm$^2$. The procedure involves adjusting brightness and contrast so that feeding appears white and the intact leaf surface appears black and then calculating feeding area based on previous calibration with a ruler image. The strongly incised leaves of *Lepidium squamatum* precluded using this method for that species so it was excluded from analysis for *P. xylostella*.

## Glucosinolate analysis

The leaves of the other five individual plants from each species, and from the second populations of six of the test species (Table 1), were collected for chemical analysis (total number of plants = 95). These leaves were collected at the same time as the bioassays, and following the same blocking structure used for bioassay (A–E). Since the location and age of tissues could affect GS profiles, old and young leaves were sampled separately on each plant (five old leaves and five young leaves per plant species and/or population) to be analyzed separately for GS content. Younger leaves were closer in age to those used in the bioassay than older leaves, and were above the middle node of the plant. Older leaves were from below the middle node of the plant. To prevent enzymatic hydrolysis of GS, the leaves were immediately placed in liquid nitrogen and stored at –40 °C before being transported frozen to the University of Bern where they were lyophilized for 24 h. The dried leaf material was then shipped to the University of Idaho for chemical analyses.

Analysis of GS profiles in leaf tissue was conducted using an established method (*International Organization of Standardization (ISO), 1992*) as modified by *Borek & Morra (2005)*. First, the lyophilized leaf material was ground. Up to 100 mg of ground tissue was placed into 14-ml polypropylene centrifuge tube with 10 to 15 3-mm glass beads, 12 mL of a 70% methanol/water solution and 200 µL of a 10 µMol/L solution of the internal standard (IS), 4-methoxybenzyl GS. The IS was obtained from meadow foam (*Limnanthes alba*) seed meal (*Hanley, Heaney & Fenwick, 1983*) and was absent from the GS profiles of all test species. The tube was capped, vortexed thoroughly, and agitated for 2 h on a shaker. The tube was then centrifuged for 5 min at 1,073 $g_0$, and the supernatant was filtered under vacuum using a Whatman 0.2 µm size pore NYL W/GMF filter. The filtered supernatant was placed onto a DEAE anion exchange column (250 mg Sephadex A 25) which was then rinsed twice with 1 mL of deionized water. The trapped GS were desulfated by sequentially rinsing the column with 1.5 mL of 0.1M of ammonium acetate buffer (pH 4.0), 200 µL of a 1 mg/L solution of sulfatase (Sigma-Aldrich, St. Louis, MO, USA), and another 200 µL of 0.1M of ammonium acetate buffer (pH 4.0). The columns were loosely capped overnight, after which the desulfated GS were eluted with 1 ml of boiled deionized water and stored at –20 °C until analysis.

The samples were analyzed by high performance liquid chromatography (HPLC) (1200 HPLC separation system, Agilent Technologies, Palo Alto, CA, USA) with a total-ion-current detector coupled with a time-of-flight mass spectrometer (Agilent

Technologies 6210) with an electrospray interface. Separation was performed on a 250 × 2.00 mm, 5 μm, 125 Å Aqua C18 column (Phenomenex, Torrance, CA, USA). The mobile phase was a methanol gradient starting at 0.5% and increasing to 50%, with a flow rate of 200 μL/min. The desulfated GS were visualized with an extracted ion chromatogram (m/z values of 195.00000–196.50000, bracketing the m/z of the glucose-S moiety, m/z = 195.0327). Compounds containing the glucose-S moiety were identified as GS and further identified based on the M−1, M+45, M+113, M+196, 2M−1 and 2M−1 fragments.

A total of 45 GS compounds were detected (Table S2) and of these 26 were selected for statistical analysis, as follows. Six of the original 45 were discarded because they could not be clearly identified as GS. Six others were collapsed into three based on indistinguishable mass spectra and near coelution (e.g., 4-(methylsulfonyl)butyl GS and 4-(methylsulfinyl)butyl GS, were combined as were 9-(methylsulfonyl)nonyl GS and 9-(methylsulfinyl)nonyl GS). All five compounds eluting after 30 min and identified as GS could not be more precisely identified and were discarded as likely dimers.

## Genetic phylograms and glucosinolate dendrograms

Plant leaf samples for a phylogenetic analysis were collected as part of an earlier study (H. Hinz, 2006, unpublished data). Genomic DNA was extracted from approximately 20 mg of silica dried leaf material using a modified cetyltrimethyl ammonium bromide (CTAB) method (Hillis et al., 1996). The chloroplast gene *ndhF* was PCR (Polymerase Chain Reaction) amplified from genomic DNA and sequenced using the same primers as in Beilstein, Al-Shehbaz & Kellogg (2006). DNA sequencing, using an Applied Biosystems (Foster City, CA, USA) 3130 Genetic Analyzer, was performed as in Gaskin, Zhang & Bon (2005). DNA sequences were aligned using CLUSTAL W (Thompson, Higgins & Gibson, 1994) in MEGA7 (Kumar, Stecher & Tamura, 2016). Nucleotide positions containing gaps and missing data were eliminated. There were a total of 1769/2046 base-pair positions used in the final dataset. Evolutionary distances were computed from chloroplast gene *ndhF* sequences using the Maximum Composite Likelihood model (Tamura, Nei & Kumar, 2004). Evolutionary analyses were also conducted in MEGA7. The phylogram was inferred using the Maximum Likelihood method based on the General Time Reversible model (Nei & Kumar, 2000) with bootstrap values (1000 replicates) shown next to the branches of the generated tree. The tree was drawn to scale, with branch lengths in the same units as those of the evolutionary distances used to infer the phylogenetic tree. Phenotypic dendrograms based on concentrations of individual GS for the 13 plant species for *C. cardariae* (and 12 species for *P. xylostella*, leaving out *L. squamatum*) were generated using a neighbor-joining algorithm. Distances were computed using the default method for the function dist() in the Core Stats package v3.6.1 (R Core Team, 2013). Trees were generated from the nj() function in the package ape v5.3. Bootstrap values for the trees were computed using the boot.phylo() function in package ape v5.3 with 100 interations. Separate dendrograms were constructed using GS data from young leaves, from old leaves, and for the mean values of both. When two populations

were assessed for GS within a species, the mean GS concentrations for that species were used as inputs.

## Statistical analysis

The effect of plant species on total GS was assessed using analysis of variance (ANOVA) on the untransformed total GS for each plant ($n$ = 5 for those species represented by a single population and $n$ = 10 for those with two populations), which conformed to normality assumptions. Although the effect of population could not be tested in the whole model because only six species had two populations, single degree-of-freedom contrasts compared total GS between population in each of those six species. The number of feeding punctures produced by *C. cardariae* females was compared with a Kruskal–Wallis nonparametric test due to nonnormality and zero values for two of the species. The effect of plant species on leaf area removed (mm$^2$) by *P. xylostella* was compared with ANOVA on the untransformed values, which conformed to normality assumptions. The effect of temporal block was not significant ($P$ > 0.05) for either analysis (ANOVA and Kruskal–Wallis) and was excluded from the final tests.

As a first test of the hypothesis that, for *C. cardariae* and *P. xylostella* separately, chemical profiles better predict insect feeding than phylogenetic relatedness, we evaluated feeding intensity as a function of genetic similarity to *L. draba*, chemical similarity to *L. draba*, and total GS in leaf tissue. Multiple linear regressions were then calculated to predict feeding by each species based on the independent variables. In addition, correlations between feeding intensity by each insect species and each individual GS were calculated. Due to the large number of zero values for concentrations in the dataset, Spearman's rank correlation was estimated. Analyses of total GS and of associations between feeding, genetic distance, GS similarities and individual GS concentrations were calculated in JMP 13.2 (SAS Institute, Cary, NC, USA).

As a corroborating and more comprehensive comparison of the relative strength of genetic relatedness and similarity of GS profiles in determining feeding intensity by each insect species, the associations between feeding, either with genetic phylograms or dendrograms based on GS profiles were examined with the R package "PhyloSignal" ((*Keck et al., 2016*) in R version 3.5.1 (Team 2013)). PhyloSignal estimates whether related species resemble each other in terms of some continuous trait more than species drawn at random from the same tree. The package computes an estimate of "kappa," or Blomberg's $K$ (*Blomberg, Garland & Ives, 2003*)—a value for $K$ of 1 indicates relatives resemble one another as expected under Brownian motion, values <1 indicate a phylogenetic signal less than expected while $K$ > 1 indicates greater association with a phylogram or dendrogram—and computes a permutation test to assign a $P$ value to the strength of the association. We calculated Blomberg's $K$ for feeding intensity using both our phylogeny of the test species and the chemical, phenotypic dendrograms for these species. We compared these $K$ values using a resampling method with 1,000 iterations.

| Glucosinolate name | LD | LD2 | LC | LC2 | LS | LS2 | LCR | LL | LL2 | BO | CM | CM2 | HM | SP | BN | BN2 | CI | DN | SV |
|---|---|---|---|---|---|---|---|---|---|---|---|---|---|---|---|---|---|---|---|
| 3-(methylsulfinyl)propyl | 0 | 0 | 0 | 0 | 0 | 0 | 0 | 0 | 0 | 0 | 0 | 0 | 0 | 0 | 0 | 0 | 6.57 | 0 | 0 |
| 2-hydroxypropyl | 0 | 0 | 0 | 0 | 0 | 0 | 0 | 0 | 0 | 0 | 0 | 0 | 0 | 11.69 | 0 | 0 | 0 | 0 | 0 |
| 2-(R)-hydroxy-3-butenyl or 2-(S)-hydroxy-3-butenyl | 0 | 0 | 0 | 0 | 0 | 0 | 0 | 0 | 0 | 0 | 0 | 0 | 0 | 0 | 0 | 0 | 0 | 0 | 0.89 |
| 2-(R)-hydroxy-3-butenyl or 2-(S)-hydroxy-3-butenyl | 0 | 0 | 0 | 0 | 0 | 0 | 0 | 0 | 0 | 0 | 0 | 0 | 0 | 0 | 0 | 0 | 0 | 0 | 0.61 |
| 2-propenyl | 0 | 0 | 0 | 0 | 0 | 0 | 0 | 20.76 | 3.6 | 0 | 0 | 0 | 0 | 0 | 18.56 | 25.07 | 0.42 | 0.25 | 1.34 |
| 4-(methylsulfonyl)butyl and 4-(methylsulfinyl)butyl | 20.28 | 17.31 | 0.04 | 0.94 | 0 | 0 | 0 | 0 | 0 | 0 | 0 | 0 | 0 | 0.72 | 0.04 | 0 | 0 | 0 | 0.09 |
| 2-hydroxy-4-pentenyl | 0 | 0 | 0 | 0 | 0 | 0 | 0 | 0 | 0.31 | 0 | 0 | 0 | 0 | 0 | 0 | 0 | 0 | 0 | 0 |
| propyl | 0 | 0 | 0 | 0 | 0 | 0 | 9.97 | 0 | 0 | 0 | 0 | 0 | 0 | 21.42 | 0 | 0 | 0 | 0 | 5.51 |
| 4-hydroxybenzyl | 19.15 | 12.45 | 20.89 | 20.91 | 1.36 | 1.03 | 1.08 | 0.87 | 1 | 0.67 | 0.79 | 0.77 | 2.55 | 1.45 | 0.97 | 0.86 | 0.87 | 1.06 | 1.53 |
| 5-(methylsulfinyl)pentyl | 0 | 0 | 4.9 | 5.95 | 0 | 0 | 0 | 0 | 0.2 | 0 | 0 | 0 | 0 | 0 | 0 | 0 | 0 | 0 | 0 |
| 3-butenyl | 0 | 0 | 0 | 0 | 0 | 0 | 0 | 0 | 7.49 | 0 | 0 | 0 | 0 | 0 | 0 | 0.04 | 0 | 0 | 10.76 |
| 2-hydroxybenzyl | 0 | 0 | 0 | 0 | 0.06 | 0 | 0.16 | 0 | 0 | 0 | 0 | 0 | 0 | 0 | 0 | 0 | 0 | 0 | 0 |
| 4-hydroxyindol-3-ylmethyl | 0.29 | 0.17 | 0 | 0.02 | 0 | 0 | 0 | 0.04 | 0.09 | 0 | 0 | 0 | 1.25 | 0.76 | 1.02 | 1.46 | 0.62 | 1.91 | 0.34 |
| unknown | 0 | 0 | 0 | 0 | 1.17 | 0.73 | 0 | 0 | 0 | 0 | 0 | 0 | 0 | 0 | 0 | 0 | 0 | 0 | 0 |
| 3,4-dihydroxybenzyl | 0 | 0 | 0 | 0 | 1.03 | 0.35 | 0 | 0 | 0 | 0 | 0 | 0 | 0 | 0 | 0 | 0 | 0 | 0 | 0 |
| 6-(methylsulfinyl)hexyl | 0 | 0 | 0.34 | 0.43 | 0 | 0 | 0 | 0 | 0 | 0 | 0 | 0 | 0 | 0 | 0 | 0 | 0 | 0 | 0 |
| 2-methoxybenzyl and 3-methoxybenzyl | 0 | 0 | 0 | 0 | 0 | 0 | 0 | 0 | 0 | 51.22 | 0 | 0 | 0 | 0 | 0 | 0 | 0 | 0 | 0 |
| butenyl | 0 | 0 | 0 | 0 | 0 | 0 | 27.03 | 0 | 0 | 0 | 0 | 0 | 0 | 1.82 | 0 | 0 | 0 | 0 | 4.27 |
| benzyl | 0 | 0 | 0 | 0 | 0.04 | 0 | 0.74 | 6.99 | 20.76 | 0 | 0 | 0 | 0 | 0 | 0 | 0 | 0.31 | 0 | 1.12 |
| 4-methylthiobutyl | 1.02 | 0 | 0 | 0.06 | 0 | 0 | 0 | 0 | 0 | 0 | 0 | 0 | 0 | 0 | 0 | 0 | 0 | 0 | 0 |
| 3,4-dimethoxybenzyl | 0 | 0 | 0 | 0 | 0.22 | 0.3 | 0 | 0 | 0 | 0 | 0 | 0 | 0 | 0 | 0 | 0 | 0 | 0 | 0 |
| indolyl-methyl | 0.06 | 0.08 | 0 | 0.11 | 0 | 0 | 0 | 0 | 0 | 8.7 | 0 | 0 | 0 | 0 | 0.54 | 0.83 | 0.22 | 0 | 1.87 |
| unknown | 0 | 0 | 0 | 0 | 8.61 | 9.22 | 0 | 0 | 0 | 0 | 0 | 0 | 0 | 0 | 0 | 0 | 0 | 0 | 0 |
| 2-phenylethyl | 0 | 0 | 0 | 0 | 0 | 0 | 0 | 0 | 0 | 7.6 | 0 | 0 | 0 | 0 | 0.4 | 2.52 | 0 | 0 | 0 |
| 4-methoxyindol-3-ylmethyl | 1.17 | 0.57 | 0.84 | 1.51 | 0 | 0.06 | 0.15 | 0.84 | 1.14 | 0 | 0 | 0 | 0.26 | 0 | 0 | 0 | 0.33 | 1.58 | 1.19 |
| 9-(methylsulfonyl)nonyl and 9-(methylsulfinyl)nonyl | 0 | 0 | 0 | 0 | 0 | 0 | 0 | 0 | 0 | 0 | 0 | 0 | 0 | 0 | 0 | 0 | 0 | 11.84 | 0 |
| **Total GS** | 41.97 | 30.58 | 27.01 | 29.93 | 12.49 | 11.69 | 39.13 | 29.5 | 34.59 | 68.19 | 0.79 | 0.77 | 4.78 | 37.18 | 21.49 | 30.78 | 9.34 | 16.64 | 29.52 |

Legend — µmol/g: 0, 1, 2, 3, 4, 5, 6, 7, 8, 9, 10, 20, 30, 40, 50

**Figure 1** **Mean concentrations (µmol/g) in leaf tissue for each of 26 GS detected in the test species used for analysis.** Plant species are sorted in order of their genetic distance from *Lepidium draba*. Species names in order, left to right: LD, *Lepidium draba*; LC, *Lepidium campestre*; LS, *L. squamatum*; LCR, *L. crenatum*; LL, *L. latifolium*; BO, *Barbarea orthoceras*; CM, *Camelina microcarpa*; HM, *Hesperis matronalis*; SP, *Stanleya pinnata*; BN, *Brassica nigra*; CI, *Caulanthus inflatus*; DN, *Draba nemorosa*; SV, *Stanleya viridiflora*; if a second population was analyzed, it is indicated with a number, for example, LL2 = the second population of *L. latifolium*. For calculating the dendrogram based on GS, mean values were used for these species. Legend: fill color visualizes the µmol GS/g.

## RESULTS

A summary of the GS profiles using the average values for samples of young and old leaves of the test species, and for multiple populations when included, is presented in Fig. 1. To visualize the similarity of the GS profiles for each species with *L. draba*, the profiles in Fig. 1 are in a tabular format with species sorted in order of their genetic distance from *L. draba* and with mean concentrations represented by color intensity. Total GS concentration differed among species (Table 2). The effect of population nested within species could not be tested in the whole model because most species were represented by a single population. Based on individual contrasts comparing populations within each species only those for *L. draba* differed in total GS (*t*-ratio = 2.094; $P = 0.0378$), and not for other species with two populations (*t*-ratio absolute value $<1.5$, $P > 0.10$). Feeding punctures by *C. cardariae* differed among the 13 species tested in bioassays ($\chi^2 = 45.7$; $P < 0.0001$) with greatest feeding occurring on *L. draba* followed by feeding on the closely related European *L. campestre* and the North American *D. nemorosa* (Table 3). Little to no feeding occurred on four plant species. Leaf area removed by *P. xylostella* did not differ significantly among the 12 species tested (ANOVA $F_{12,50} = 1.57$; $P = 0.130$) (Table 3).

Feeding intensities by *C. cardariae* and *P. xylostella* on the test plant species exhibited non-significant negative trends with phylogenetic distance from *L. draba* and GS profile distance from *L. draba*, and positive trends with total GS in leaf tissue (Fig. 2). Among

**Table 2 Total GS concentrations in each plant species and 2nd populations (if tested) of each species used in this study.** Total GS concentrations in each plant species and 2nd populations (if tested) of each species used in this study (mean ± SE, $n$ = 5).

| Plant species (population) | Total GS (µmol/g) |
|---|---|
| *Barbarea orthoceras* | 68.2 ± 5.6 |
| *Brassica nigra* | 21.5 ± 2.9 |
| *Brassica nigra* (2nd pop) | 30.8 ± 2.8 NS |
| *Camelina microcarpa* | 0.8 ± 0.1 |
| *Camelina microcarpa* (2nd pop) | 0.8 ± 0.0 NS |
| *Caulanthus inflatus* | 9.3 ± 2.1 |
| *Draba nemorosa* | 16.6 ± 2.3 |
| *Hesperis matronalis* | 4.8 ± 1.1 |
| *Lepidium campestre* | 27.0 ± 2.5 |
| *Lepidium campestre* (2nd pop) | 29.9 ± 1.9 NS |
| *Lepidium crenatum* | 39.1 ± 4.8 |
| *Lepidium draba* | 42.0 ± 7.2 |
| *Lepidium draba* (2nd pop) | 30.6 ± 4.5* |
| *Lepidium latifolium* | 29.5 ± 2.6 |
| *Lepidium latifolium* (2nd pop) | 34.6 ± 7.5 NS |
| *Lepidium squamatum* | 12.5 ± 1.3 |
| *Lepidium squamatum* (2nd pop) | 11.7 ± 1.1 NS |
| *Stanleya pinnata* | 37.2 ± 9.0 |
| *Stanleya viridiflora* | 29.5 ± 6.0 |
| ANOVA $F_{df}$ for effect of species | $23.0323_{12,164}$ |
| ANOVA $P > F$ for effect of species | <0.0001 |

**Notes:**
* Populations differ.
NS = populations do not differ (LS Means contrast, $P < 0.05$).

these, only the regression between *C. cardariae* feeding and GS profile distance was significant (Fig. 2). A multiple regression for *C. cardariae* feeding using all three of these independent variables was significant (Table 4). The effect of GS profile distance was significant but not phylogenetic distance or total GS. The same model for *P. xylostella* was not significant overall or for any effect (Table 4).

Spearman rank correlations between *C. cardariae* feeding and individual GS were significant ($P < 0.05$) for propyl GS and butenyl (both –0.570) and 4-methylthiobutyl (0.631). For *P. xylostella* feeding, none were significant.

Dendrograms for phenotypic similarity were based on all 26 GS listed in Fig. 1 for *C. cardariae*, and 24 GS for *P. xylostella* (since GS 14 and 15 were only detected in *L. squamatum*, for which bioassay data could not be acquired with our method for *P. xylostella*). GS data for old and young leaves separately (Tables S3 and S4) were also used to generate dendrograms. Dendrograms based on GS from young leaves, old leaves or all leaves differed slightly (Fig. S1) and mean total GS concentrations were higher for younger leaves than for older leaves (33.4 ± 6.2 vs. 17.8 ± 4.1 µmol/g, young vs. old), but overall patterns and conclusions are the same (Table 5).

**Table 3 Feeding intensity by *C. cardariae* (number of feeding punctures) and *P. xylostella*.** Feeding intensity by *C. cardariae* (number of feeding punctures) and *P. xylostella* (leaf area removed in mm²) (mean ± SE, $n = 5$) on plant species in the family Brassicaceae used in this study.

| Plant species | Nativity | *C. cardariae* # feeding punctures | *P. xylostella* leaf area removed (mm²) |
|---|---|---|---|
| *Barbarea orthoceras* | NA | 13.4 ± 6.8 | 14.0 ± 4.2 |
| *Brassica nigra* | EU | 26.0 ± 5.9 | 24.0 ± 4.7 |
| *Camelina microcarpa* | NA | 0.0 | 10.6 ± 4.1 |
| *Caulanthus inflatus* | NA | 23.6 ± 12.4 | 15.2 ± 3.4 |
| *Draba nemorosa* | NA | 46.6 ± 12.7 | 13.4 ± 4.1 |
| *Hesperis matronalis* | NA | 3.4 ± 2.0 | 10.8 ± 1.7 |
| *Lepidium campestre* | EU | 65.8 ± 29.2 | 12.7 ± 3.3 |
| *Lepidium crenatum* | NA | 3.4 ± 2.0 | 11.6 ± 4.1 |
| *Lepidium draba* | NA | 125.0 ± 35.5 | 19.3 ± 2.1 |
| *Lepidium latifolium* | EU | 17.6 ± 4.5 | 19.8 ± 3.4 |
| *Lepidium squamatum* | EU | 9.5 ± 0.5 | N/A |
| *Stanleya pinnata* | NA | 1.2 ± 0.4 | 13.0 ± 1.6 |
| *Stanleya viridiflora* | NA | 0.0 | 9.3 ± 1.7 |
| ANOVA $F_{df}$ | | – | $1.68_{11,46}$ |
| ANOVA prob. > $F$ | | – | 0.1080 |
| Kruskal–Wallis approx. $\chi^2$ | | 42.260 | – |
| Kruskal–Wallis prob. > $\chi^2$ | | <0.0001 | – |

Although associations between the phylograms and feeding intensity based on the value of $K$ were not significant ($P$ value > 0.05) for either species (Fig. 3; Table 5), the values of $K$ were higher and the $P$ values were lower for associations of feeding with the dendrogram based on GS similarity than with the phylogram. For *P. xylostella*, relatively high feeding intensity on several *Lepidium* spp. and on *Brassica nigra*, which are distantly related genetically but more similar in GS profiles, contributed to this pattern.

For *C. cardariae*, the relative strength of the association with the GS dendrogram is much stronger than that with the phylogram based on genetic distance. The reason for this is harder to discern from the figures than the difference for *P. xylostella* but is reflected in the more similar feeding intensity on pairs of plants that fall into the same clusters, notably *L. latifolium* and *Brassica nigra* (Fig. 3).

For *C. cardariae*, associations between feeding intensity and a dendrogram based on GS profiles of old leaves (based on $K$ and $P$ values) was similar to that for the dendrogram based on GS for all leaves, while the association with a dendrogram based on GS from young leaves was the strongest detected for this species in this study (Table 5).

For *P. xylostella*, the association between feeding intensity and a dendrogram based on GS profiles of old leaves was the weakest, while the association between feeding intensity and the dendrogram based on GS of young leaves was similar to the association with the dendrogram based on all GS (Table 5).

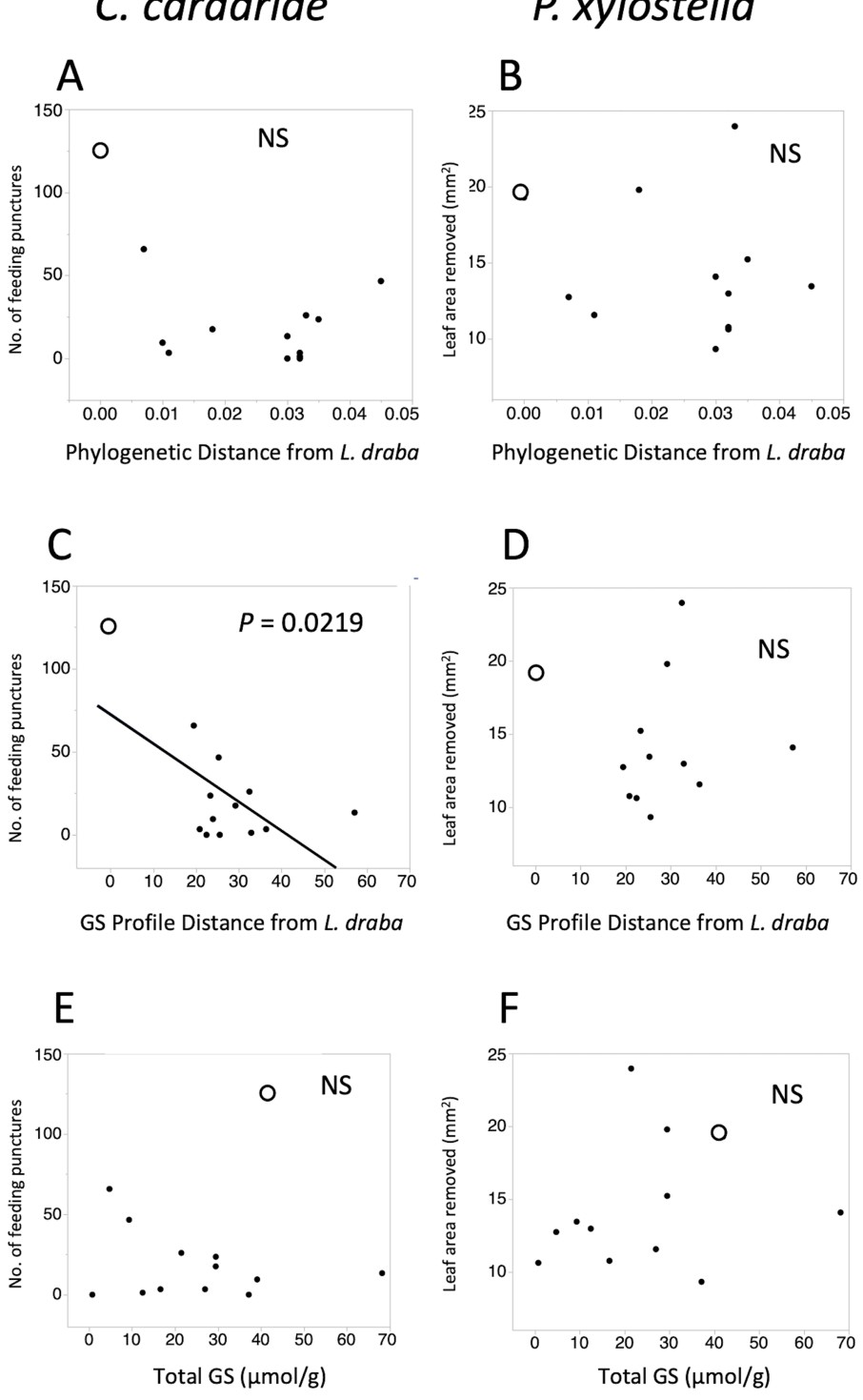

**Figure 2 Scatter plots of feeding intensity by *C. cardariae* (feeding punctures) and *P. xylostella* (mm² removed) in bioassays.** Scatter plots of feeding intensity by *C. cardariae* (feeding punctures) and *P. xylostella* (mm² removed) in bioassays, plotted against (A, B) the phylogenetic distance, (C, D) GS profile distance from *L. draba* based on nearest neighbor joining algorithm with 26 GS or (E, F) total GS concentration in leaf tissue. Open points are *L. draba* in each panel. Regressions were calculated for each and the *P* value is provided when significant.

**Table 4 ANOVA table for multiple regressions of feeding by *C. cardariae* and *P. xylostella*.** ANOVA table for multiple regressions of feeding by *C. cardariae* and *P. xylostella* in response to phylogenetic distance of each test species from *L. draba*, GS profile distance from *L. draba*, and total GS.

| Insect species | Effect[*] | SS | F | P > F[**] |
|---|---|---|---|---|
| *C. cardariae* | Model | 9510.67 | 4.9264 | 0.0271 |
| | Phylogenetic distance | 62.77 | 0.0975 | 0.7619 |
| | GS profile distance | 5630.92 | 8.7502 | **0.0160** |
| | Total GS | 732.21 | 1.1378 | 0.3139 |
| *P. xylostella* | Model | 19.59 | 0.2711 | 0.8447 |
| | Phylogenetic distance | 0.11 | 0.9471 | 0.9471 |
| | GS profile distance | 6.25 | 0.2594 | 0.6243 |
| | Total GS | 11.59 | 0.4808 | 0.5077 |

Notes:
[*] df: Model = 3, Phylogenetic distance, GS profile distance, Total GS = 1; for *C. cardariae*, error = 9, for *P. xylostella*, error = 8.
[**] Values in boldface are significant, below the standard threshold of 0.05.

**Table 5 Mean *K* values for the association of the feeding intensity by *C. cardariae* and *P. xylostella*.** Mean values of *K* values for the association of the feeding intensity by *C. cardariae* and *P. xylostella* mapped onto phylograms (trees) based on genetic similarity (chloroplast gene *ndh*F. or GS profiles (concentration in μmol/g) of the plant species tested in bioassays. A value of $K = 0$ indicates no association between the phylogram or dendrogram, while higher values indicate an association is indicated.

| Species Tree type | K value | P-value |
|---|---|---|
| *C. cardariae* | | |
| Genetic distance, phylogram | 0.28 | 0.4411 |
| GS, dendrogram, all leaves | 0.91 | 0.0797 |
| GS, dendrogram, old leaves | 0.84 | 0.1202 |
| GS, dendrogram, young leaves | 1.38 | 0.0279 |
| *P. xylostella* | | |
| Genetic distance, phylogram | 0.75 | 0.0910 |
| GS dendrogram, all leaves | 0.95 | 0.0710 |
| GS, dendrogram, old leaves | 0.73 | 0.0858 |

# DISCUSSION

Although the host ranges of insect herbivores generally are phylogenetically constrained (*Bernays & Chapman, 1994*; *Rasmann & Agrawal, 2011*), some species have phylogenetically disjunct host ranges that include plant species more distantly related to a primary host but exclude some that are more closely related (*Hinz & Diaconu, 2015*; *Weiblen et al., 2006*). Plant secondary metabolites that are not phylogenetically conserved among species can explain phylogenetically disjunct patterns of insect host use (*Becerra, 1997*; *Becerra, Noge & Venable, 2009*; *Pearse & Hipp, 2009*). In the present study, we show that feeding intensity by *C. cardariae* is more strongly related to the similarity of the GS profiles of its potential hosts to the GS profile of *L. draba*, its field host in its native range, than to the relatedness of these species to *L. draba*, helping to explain its disjunct host

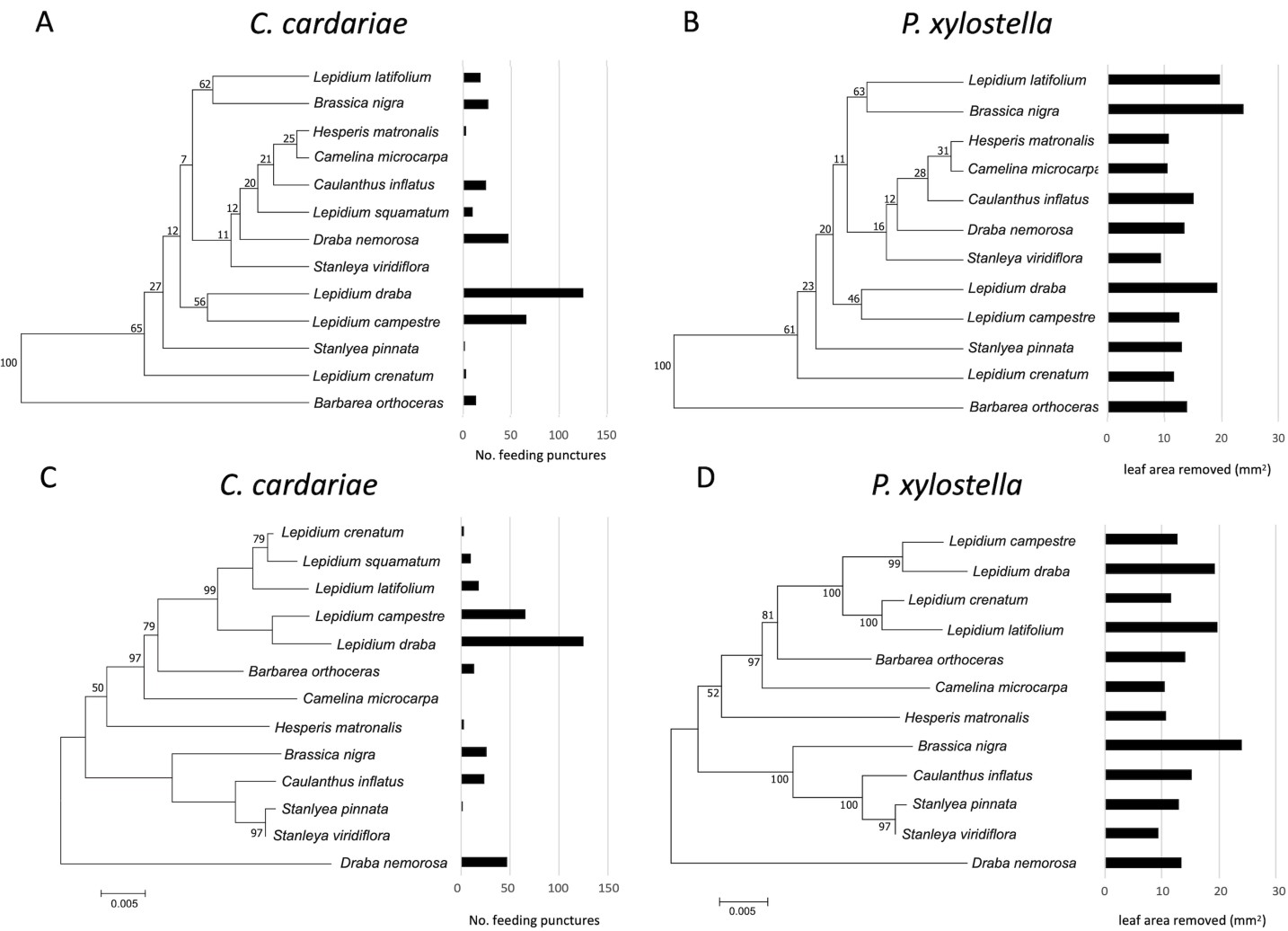

**Figure 3 Mean number of feeding punctures by *C. cardariae* and leaf area removed by *P. xylostella* in bioassays mapped onto dendrograms based on the similarity in GS concentrations and phylograms based on genetic similarity among the test.** Mean number of feeding punctures by *C. cardariae* and leaf area removed (mm²) by *P. xylostella* in bioassays mapped onto (A and B, respectively) dendrograms based on the similarity in GS concentrations (μmol GS/g) (mean of young and old leaves) among tested species, and (C and D, respectively) phylograms based on genetic similarity among the test species. For *C. cardariae* (A and C), 13 species are included while for *P. xylostella* (B and D) 12 species are included. The scale bar beneath the phylograms indicates the number of base substitutions per site between sequences.

range. GS are the focus here based on their roles as chemical defenses against generalists and feeding stimulants for more specialized insect herbivores attacking Brassicaceae (*Hambäck et al., 2009*; *Schoonhoven, Van Loon & Dicke, 2005*), and the phylogenetic lability of GS profiles (*Arany et al., 2008*; *Bidart-Bouzat & Kliebenstein, 2008*; *Gols et al., 2008*; *Olsen et al., 2016*). Feeding by *C. cardariae* is not related to total GS concentration, but to overall GS profiles, consistent with the widely varying biological activity of individual GS (*Newton, Bullock & Hodgson, 2009*; *Hopkins, Van Dam & Van Loon, 2009*; *Robin et al., 2017*). Although three individual GS were correlated with *C. cardariae* feeding (propyl GS, butenyl, and 4-methylthiobutyl), the signs of these correlations differed and it is difficult to assign biological meaning with this dataset.

In contrast to *C. cardariae*, *P. xylostella* exhibited no significant associations between feeding intensity and either GS or phylogenetic similarity of hosts. *Plutella xylostella* feeding did not differ as strongly among these species as did feeding by *C. cardariae* (e.g., the range of normalized feeding intensities (values/overall mean from Table 3) for *C. cardariae* (1.82), twice that of *P. xylostella* (0.98)). This is consistent with the relatively broad host range of *P. xylostella* (*Sarfraz et al., 2011*; *Shelton & Nault, 2004*; *Talekar & Shelton, 1993*, CABI—https://www.cabi.org/isc/datasheet/42318). Tests with more potential hosts than those tested here, which were selected specifically for *C. cardariae*, might detect associations between *P. xylostella* feeding intensity and phenotypic and phylogenetic similarity.

Although *C. cardariae* feeding is more strongly associated with GS similarity than phylogenetic relatedness of its potential hosts, that association is relatively weak. Inspection of Figs. 2 and 3 reveals inconsistencies that contribute to those weak associations: the variation around the fitted line in Fig. 2 and the wide range of feeding intensities for *C. cardariae* within a phenotypic clade in Fig. 3. This variation might be reduced by including more phenotypic traits. Candidates include other secondary metabolites such as flavonoids (*Van Loon et al., 2002*), triterpenoids (*Agerbirk et al., 2003*; *Eigenbrode & Pillai, 1998*), physical traits including leaf toughness (*Hariprasad & van Emden, 2010*), pubescence (*Gupta & Thorsteinson, 1960*), plant volatiles and surface waxes (*Sarfraz, Dosdall & Keddie, 2006*), and saponins (*Shinoda et al., 2002*) all of which can influence *P. xylostella* feeding or oviposition, and potentially also affect *C. cardariae*. The effects of GS profiles could depend upon myrosinase enzyme activity, which can vary among genotypes within a species and likely does among species in Brassicaceae, with implications for mobilization of GS hydrolysis products for defense against specialists and generalists (*Li et al., 2000*). Plant volatiles and surface chemistry have been implicated in influencing *C. cardariae* feeding (*Rendon, 2019*) and could be included. *Stanleya* spp. can accumulate selenium that can impart defense (*Freeman et al., 2012*). In addition to secondary metabolites, host species may differ in primary metabolite profiles such as sugars and amino acids that contribute to differential feeding. Including characterization of these plant traits could help explain host use by *P. xylostella* and *C. cardariae*. Traits important for each species are likely to differ, however, first because *P. xylostella* has a broader host range and second because feeding strategies differ between these species: *C. cardariae* excavates deep pits prior to oviposition, while *P. xylostella* is a strip feeder in the instar tested, likely consuming more epidermis as a proportion of tissues ingested, exposing it more to the effects of leaf toughness, trichomes, and surface waxes.

*Rasmann & Agrawal (2011)* concluded that plant traits, including secondary chemistry (total cardenolide content) can explain the differences in herbivore performance associated with phylogenetic distance from a primary host (*T. tetraophthalmus* feeding on 18 *Aesclepias* spp., including the primary host *A. syriaca*). We found no associations with total GS, but only with GS profile similarity. For GS, this is consistent with the differing effects of individual GS on the physiology and behavior of insect herbivores, including specialists and generalists (*Newton, Bullock & Hodgson, 2009*; *Hopkins, Van Dam & Van Loon, 2009*; *Robin et al., 2017*).

*Ceutorhynchus cardariae* feeding was more strongly associated with a dendrogram based on GS from younger leaves than one based on GS from all leaves or from older leaves (Table 5). This may be because somewhat younger leaves were used in bioassays, although selected from the approximate middle nodes of the plant. Younger leaves are also typically preferentially fed upon by *C. cardariae* (C. Rapo, 2008, personal observations) and for some other weevil species a preference for younger leaves has been reported. Younger leaves have higher GS concentrations. Defenses, including GS, also vary among tissues and with development in individual plants (*Bingaman & Hart, 1992*; *Bingaman & Hart, 1993*; *Smith & Griffiths, 1988*; *Wallace & Eigenbrode, 2002*), with implications for oviposition and feeding (*Fei et al., 2017*; *Vaughn & Hoy, 1993*).

In addition to revealing the basis of phylogenetically disjunct host ranges of insect herbivores, this study has practical implications for prerelease testing of potential weed biological control agents. Prerelease studies require controlled bioassays to determine the risks of nontarget attack. Nontarget species for testing are typically selected based on their phylogenetic relatedness to the target under the expectation that plant traits relevant to host use by phytophagous insects are phylogenically conserved (*Briese, 2005*; *Jaenike, 1990*; *Pearse & Altermatt, 2013*; *Sheppard, Klinken & Heard, 2005*; *Wapshere, 1974*). Although typically sound (*Hinz, Winston & Schwarzländer, 2019*; *Wheeler & Madeira, 2017*) this approach could overlook potential nontarget species that are more distantly related if relevant plant traits are not phylogenetically conserved. Evidence of a phylogenetically disjunct host range in a candidate agent could indicate that metabolic profiles be used as a first step to select species for inclusion in bioassays to assess risks of nontarget attack.

## CONCLUSIONS

In this paper, we test the hypothesis that the phytochemical similarity among potential host plants better explains their relative suitability for feeding by a relatively specialized insect herbivore than does the phylogenetic relatedness of these plant species. The hypothesis was confirmed for *C. cardariae* feeding on potential hosts within its phylogenetically disjunct fundamental host range. The finding is consistent with the premise that insect host range is determined by phenotypic similarity among its hosts rather than relatedness per se. Plant traits that influence feeding, if phylogenetically labile, can produce phylogenetically disjunct host ranges. This is important because it can help explain some patterns of host use among insect hervivores. It also has practical implications for prerelease testing of potential weed biological control agents, helping to structure this testing to improve its efficiency and effectiveness for detecting risks of nontarget attack.

## ACKNOWLEDGEMENTS

We are grateful to the late Vladimir Borek for his help with GS extractions and identification. Thanks to Anurag Agrawal, Luke Harmon, Nicole van Dam, and Caroline Müller for helpful discussions during early stages of the project. Thanks to Willy Tinner (University of Bern) for the use of the lyophiliser. We are grateful to Florencia Oberli,

Laura Parsons, Cornelia Closca, Sabrina Schaller, Christian Léchenne and Florence Willemin for their technical assistance.

### Funding

Funding for this research was provided by the University of Idaho EPPN Department; USDI BLM Federal Assistance Agreements L08AC14943 and DLA080108; Wyoming Biological Control Steering Committee; Montana Weed Trust Fund through Montana State University and USDA-APHIS-PPQ-CPHST. Hariet Hinz and Urs Schaffner were supported by CABI with core financial support from its member countries. The funders had no role in study design, data collection and analysis, decision to publish, or preparation of the manuscript.

### Grant Disclosures

The following grant information was disclosed by the authors:
University of Idaho EPPN Department; USDI BLM Federal Assistance Agreements: L08AC14943 and DLA080108.
Wyoming Biological Control Steering Committee.
Montana Weed Trust Fund through Montana State University.
USDA-APHIS-PPQ-CPHST.
CABI.

### Competing Interests

The authors declare that they have no competing interests.

### Author Contributions

- Carole B. Rapo conceived and designed the experiments, performed the experiments, analyzed the data, prepared figures and/or tables, authored or reviewed drafts of the paper, approved the final draft.
- Urs Schaffner conceived and designed the experiments, analyzed the data, authored or reviewed drafts of the paper, approved the final draft.
- Sanford D. Eigenbrode conceived and designed the experiments, analyzed the data, contributed reagents/materials/analysis tools, prepared figures and/or tables, authored or reviewed drafts of the paper, approved the final draft.
- Hariet L. Hinz conceived and designed the experiments, performed the experiments, authored or reviewed drafts of the paper, approved the final draft.
- William J. Price analyzed the data, prepared figures and/or tables, approved the final draft.
- Matthew Morra contributed reagents/materials/analysis tools, approved the final draft.
- John Gaskin contributed reagents/materials/analysis tools, and authored or reviewed drafts of the paper, approved the final draft, conducted the phylogenetic analysis of test plant species.
- Mark Schwarzländer conceived and designed the experiments, authored or reviewed drafts of the paper, approved the final draft.

## Data Availability

Data are available at the Northwest Knowledge Network Data Repository (NKN) (https://www.northwestknowledge.net/home) with this specific record locator: DOI 10.7923/cmm3-tj85. Data are also available in the Supplemental Files.

## Supplemental Information

Supplemental information for this article can be found online at http://dx.doi.org/10.7717/peerj.8203#supplemental-information.

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
