# Peer review of "Feeding intensity of insect herbivores is associated more closely with key metabolite profiles than phylogenetic relatedness of their potential hosts"

_PeerJ, doi:10.7717/peerj.8203_

## Round 0.1 · original submission · Major Revisions

Dear Dr. Rapo and colleagues:

Thanks for submitting your manuscript to PeerJ. I have now received three independent reviews of your work, and as you will see, the reviewers raised some concerns about the research. Despite this, these reviewers are optimistic about your work and the potential impact it will have on research communities studying the ecology and evolution of herbivorous insects. Thus, I encourage you to revise your manuscript, accordingly, taking into account all of the concerns raised by both reviewers.

Your revision should concentrate on making the text more clear and easier to follow. Specific examples are provided by the reviewers. Please also ensure that the figures and tables are clear, and that important background information is provided. Please consider including a phylogeny (either redrawn/adapted from another work or estimated here) to accompany your discussion.

While the concerns of the reviewers are relatively minor, this is a major revision to ensure that the original reviewers have a chance to evaluate your responses to their concerns.

I look forward to seeing your revision, and thanks again for submitting your work to PeerJ.

Good luck with your revision,

-joe

·

Basic reporting

This is a clearly written paper of a well-designed and well-executed study to compare the influence of plant phylogeny versus glucosinolate composition on the host plant acceptance of an oligophagous versus a polyphagous herbivorous insect. The Introduction provides a clear context and rationale for the study. It would be helpful to include some information about the natural history of the insects, especially C. cardariae (What parts of the plant do adults feed on? What time of year? Was the experiment done using 'mature' adults from the preceding year, or young, pre-reproductive adults?).

Experimental design

Methods
The experimental design and analysis generally are appropriate. However, there do not appear to be any specific methods regarding collecting specimens, extracting DNA, and sequencing the plant material. How were the plants identified? Are there voucher specimens? Provide more description of how a similarity tree was constructed using the GS data.

5 plant samples were used for chemical analysis, which should permit the authors to describe the SE for their mean values. This would be helpful to show the reliability of the results, and may also help to explain why some compounds were detected in one set of plants, but not in another of the same species (e.g., LD vs. LD2, LC vs. LS2, LS vs. LS2).

Validity of the findings

Results
The authors should check the labeling of Fig. 3 "Glucosinolate dendrogram ... A. Genetic similarity", and "Genetic similarity ... B. Phenotypic similarity". These labels appear to be contradictory. Phenotypic trees usually indicate bootstrap levels of confidence. Adding this to these two figures would help indicate how much confidence can be put on the tree patterns.
Check spelling of C. microcarpus (Fig. 3).

It is surprising to me that H. matronalis and C. microcarpa are mapped as similar on Fig. 3B (phenotype) when the latter has 3 more GS compounds than the former. Likewise, S. pinnata and L. crenatum are mapped as similar, but differ in the presence/absence of 5-7 compounds. I encourage the authors to more clearly explain the methods used and the results obtained.

Note that Fig. 4 is not mentioned in the Results section. Are these data based on all leaves, old leaves, or young leaves? Do these "young" leaves correspond to "fully expanded leaves in the upper portion ...", which were used in the feeding experiment? Considering that only young leaves had a significant p-value in Table 3, perhaps the results in Figs. 1 and 4 should focus on the young leaves. Does C. cardariae normally feed on young leaves? If yes, then these leaves would probably be more relevant to this study.

Discussion
The plants most similar in the phenotypic tree (Fig. 4) to L. draba and L. campestre (the preferred host plants) had lower levels of feeding by C. cardariae (i.e., B. orthoceras, S. pinnata, L. crenatum) than did D. nemorosa, which is mapped as having a much less similar GS profile. Fig. 2 shows D.n. as having about the same GS distance from L. draba as at least 4 other plants, which had much lower feeding. These results appear to be contrary to the conclusion that GS phenotype predicts plant feeding level. The authors should discuss details such as these.

Visually, the genetic tree (Fig. 3) appears to explain as much of the host acceptability as the glucosinolate 'phenotypic' tree (Fig. 4). Both trees place L.d. and L.c. in the same clade, and D.n. is remote from this clade in both trees. This does not appear to fit the conclusion that the GS phenotypic traits can predict feeding preference more reliably than phylogenetic distance. Neither the phylogenetic nor the glucosinolate trees 'predict' acceptability of Draba nemorosa.

I think that the Discussion could be improved by looking more closely at the results, and less on trying to make a sweeping generalized conclusion. Can the authors say/hypothesize what compounds C. cardariae likes or does not like?

Reviewer 2 ·

Basic reporting

The manuscript “Is feeding intensity of insect herbivores associated with phylogeny or glucosinolate profiles?” tackles an important question in ecological interaction between Brassicaceae plants and their specialized herbivores. The main question in this manuscript is whether the phylogenetically disjunct host range of Ceutorhynchus cardariae can be associated with glucosinolate (GS) profiles of its host plants. This question is important since C. cardariae is an important candidate for biological control of global-invasive plants, Lepidium draba, and understanding the factor which limits its host range would be beneficial to control non-target effect in such biological control. The author performed feeding experiments (including Plutella) and genetic analysis coupled with chemical analysis. It was interesting to see that C. cardariae responded more sensitively to GS profile differences of their hosts than Plutella. However, there are several points which need to be considered before publication.

The major points of criticisms
1. The author compared the results of feeding assays of C. cardariae and P. xylostella as their one of the main results. However, these two species can be different in many ecological aspects although both of them are Brassicaceae specialist. The author should at least explain that the way of feeding can be different between weevils and caterpillar and how this difference can affect the results of your feeding assay. For example, leaf toughness or trichome density can affect differently to these two species.
2. The explained experimental setting was confusing. Please clearly state how many individuals of C. cardariae and P. xylostella the author used for feeding experiments in the Material & Methods section. I assume it would be 5 from tables and raw data sets, however, it would be more useful for readers to be explained at MM section.
3. Why the author only used females of C. cardariae for the experiments? (l. 168)
4. What is the rearing condition of C. cardariae laboratory strain (l. 118). Were they reared on plants or an artificial diet? If plants were used, which plants? How this can affect the difference you found between P. xylostella larvae which were reared on artificial diet before the feeding assay?
5. Why did the authors use p-distance to estimate genetic distance? This p-distance only shows the number of nucleotide changes and does not reflect transversion-transition differences or cases of multiple substitutions (l. 227). In addition, in the ndhF sequences used in this study, there are a number of ‘N’. How did you calculate genetic distance from these sequences? If you removed such sites, please state it. You should also explain the final sequence length (bp) you used for your analysis.
6. In the manuscript, the authors explained that GS profiles of Brassicaceae are not phylogenetically conserved evident from your data sets (l. 333-). It is difficult to say this since there is no test in this manuscript addressing this point. Furthermore, I also believe that it would be difficult to test this in this dataset since there was no solid plant phylogeny provided in this manuscript. I would suggest rephrasing this such as; GS profiles can vary even among closely related species.
7. The motivation of this research is giving important information to estimate the non-target effect of C. cardariae that has phylogenetically disjunct host range but seems to respond differently depending on the GS profiles of plants. It would be more beneficial if you can point out which GSs can be candidates that would affect the feeding behavior of C. cardariae according to your results. This information would be quite important when one would design biological control of L. draba by C. cardariae practically.


Minor comments
l. 67. There are several studies focusing on herbivore host ranges associated with plat leaf traits or phylogenies although these are more community ecology based.
Nakadai, R., & Murakami, M. (2015). Patterns of host utilisation by herbivore assemblages of the genus Caloptilia (Lepidoptera; Gracillariidae) on congeneric maple tree (Acer) species. Ecological Entomology, 40(1), 14–21. doi:10.1111/een.12148
Volf, M., Hrcek, J., Julkunen-Tiitto, R., & Novotny, V. (2015). To each its own: differential response of specialist and generalist herbivores to plant defence in willows. Journal of Animal Ecology, n/a-n/a. doi:10.1111/1365-2656.12349

l. 146 Which stage of plants did you use for the feeding experiments? What was the condition of the greenhouse for rearing?

l. 241 Please add the name of the statistical test you used to get the P value (P > 0.05).
l. 270 Maybe original 43?
l. 288 P should be italicized.
l. 343 What are these normalized feeding intensities? How did you measure these?
l. 355 What does this sentence mean? There is a recent publication about Plutella GSS. You may include this information to discuss this.
Heidel-Fischer, H. M., Kirsch, R., Reichelt, M., Ahn, S.-J., Wielsch, N., Baxter, S. W., Heckel, D. G., Vogel, H., Kroymann, J. (2019). An insect counteradaptation against host plant defenses evolved through concerted neofunctionalization. Molecular Biology and Evolution, 36(5), 930-941. doi:10.1093/molbev/msz019.
l. 357 For example which GS?
l. 360 In this paragraph you may also include some information about the plant species you used for this study and how this can affect your results. For example, Stanleya is known to be toxic by accumulating Se and Brabarea is known to have saponins.
l. 381 remove “profile” before GS
In file “peerj-38348-Rapo_et_al_SUPPLEMENTARY_TABLES.docx”
There are “3rd pop” in Lepidium latifolium. Maybe this should be 2nd?

Experimental design

Some experimental methods should be explained more clearly. See basic comments.

Validity of the findings

no comment

Additional comments

no comment

Reviewer 3 ·

Basic reporting

Clear and unambiguous, professional English used throughout. --> yes, but in my opinon, some areas require rewording for clarity.

Literature references, sufficient field background/context provided. --> Some important work that provides context to the study is missing; I provide some references in the section below.

Professional article structure, figures, tables. Raw data shared. --> Article structure seems good. Find some comments in the section below.

Raw data is not shared, but this can be easily fixed.

Self-contained with relevant results to hypotheses. --> yes

Experimental design

Original primary research within Aims and Scope of the journal. --> yes

Research question well defined, relevant & meaningful. It is stated how research fills an identified knowledge gap. --> yes

Research question --> yes

Rigorous investigation --> yes

Methods described with sufficient detail & information to replicate --> no; but easily fixable. Some suggestions provided below.

Validity of the findings

no comment

Additional comments

++++++++++ Overview
In this paper, the authors assess whether feeding intensity of a specialist and a generalist feeding on mustards is best explained by genetic distance of potential hosts to true host, glucosinolate profile distance of potential hosts to true host, or total glucosinolate concentration in potential hosts. They find support for chemical distance and glucosinolate concentration being important predictors of feeding intensity, and little if no role of genetic distance.
I find the manuscript interesting and generally well written. However, references to important pieces of literature, and some information on methods and results are lacking.
My biggest issue with this paper is the use of genetic distance phylograms instead of trees derived from proper phylogenetic analyses, which could be easily, inferred using the available genetic information (or derived from the work of others). Today phylogenetic analyses are common; literature and resources to conduct such analyses is plentiful, as is the literature on comparative methods. While this implies that some downstream analyses would need to be repeated, it might be worthwhile given the strong emphasis of this paper on phylogenetic relatedness.
Below and in an accompanying PDF I offer detailed comments and suggestions as to how, in my opinion, some of the issues can be addressed.

+++++ Detailed comments

INTRODUCTION
L 65-66. In Asclepias, phylogenetic signal varies in the many defensive compounds it produces. While stronger for cardenolides, it is moderate for flavonoids, and low for phenolics (Agrawal etal., 2009). In fact, weak phylogenetic signal in secondary metabolites associated with defense has been reported in various systems, including terpenes in Bursera (Becerra etal., 2009), tannins in oaks (Pearse & Hipp, 2009), flavonoids in evening primroses (Johnson etal., 2014), phenolics in Inga, alkaloids in Datura (Karinho-Betancourt et al., 2015), and importantly for this paper, in glucosinolates in Streptanthoid mustards (Cacho et al., 2105).
Given the above, while I agree that “total cardenolide concentration in Asclepias is a factor accounting for the range of T. tetraphthalmus” I find complicated to qualify it as a “phylogenetically constrained” host range. Evidence is that phylogeny plays a rather weak role in defensive secondary compounds. I think this is the case as well in Bursera. In my opinion, including this discussion and references builds on the discussion of phylogeny vs. chemical similarity in predicting host ranges, and thus provides an important context to this study.
L 70-83. Classic examples of this very phenomenon in Bursera (work by Becerra and collaborators), which should be addressed here.
L 89-91. Glucosinolates have been shown to exhibit low phylogenetic signal (work of Cacho and collaborators) should be addressed here.
L99-102. Given the above [and your results], this is quite surprising. Chemical similarity and not phylogenetic relatedness should be the standard for this practice.
L 107-114. I do not follow the rationale under which the expectations are the same for a specialist and a generalist herbivore. If specialization does not play a significant role in the hypothesis then maybe this should be clarified and less emphasis placed on it.

METHODS
Insect section. Please make clear that female weevils are being used (and why).
L 168. Need to explain why where weevil females starved but Lepidoptera larvae were not.
L. 178. Comma between image and which is not needed; please remove.
L 192 -193. A justification or reference is needed for the criteria used to classify leaves in this way. Are both young and old leaves cauline or are “old” leaves a mix of cauline and basal leaves? Please clarify. Also, provide details on how were nodes counted in plants that have basal rosettes?
Genetic phylograms and glucosinolate dendrograms–
Given the strong emphasis put on phylogeny in this paper, I am surprised that not proper phylogenetic analyses are used in this paper. Please explain/justify why this choice was made.
Regardless of the choice of method, the alignment methods and parameters need to be included.
The Genbank numbers for ndhF sequences are provided, but in needs to be explicit if these sequences were downloaded from Genbank or generated for this study.
If proper phylogenetic analyses were not implemented, then all references to “phylogenetic relatedness” should be replaced with “genetic similarity” or something similar. The same is true for figures. References to “phylogenetic” should be replaced with “genetic”.
L 2454 – 247. I find this confusing. Here is a suggestion: For C. cardariae and P. xylostella separately, we evaluated feeding intensity as a function of genetic similarity to L. draba, chemical similarity to L. draba, and total GS in leaf tissue.
Please provide details on how were genetic and chemical distances to L. draba calculated.
Were differences between young and old leaves significant? If so, I expect separate analyses for these two different kinds of leaves.
Were assumptions of normality tested before linear models?
L. 248-260. This is an interesting idea (although the writing is a bit confusing). Phylogenetic signal is the tendency of related species to resemble each other more than species drawn at random from the same tree. Assessing whether mathematically the methods to measure degree of resemblance by different tips of a dendrogram is beyond my expertise. Has anyone implemented this approach? If so, please cite.
Suggestion: “… can work with dendrograms based on other information…” should read “…can work with dendrograms.” Please provide the number of permutations to test significance.
As far as I know, kappa is not a measure of phylogenetic signal. Literature is vast on the topic.
L. 257-259. I find this confusing. Here is a suggestion: “We calculated Blomberg’s K for feeding intensity using our genetic and chemical dendrograms in lieu of phylogenies.”
Please provide details on libraries and functions for analyses done in R.

RESULTS
L 269-270. How were the 26/43 glucosinolates selected to be retained for analyses? Please explain criteria (if these are different from only curating the GS results, see below).
L. 270. Were there 43 or 45 GS or compounds?
L 270-275. These should be in methods as GS post-HPLC curation.
L. 285-288. I had not understood the multiple regression part from the methods section. Please provide a table with the full model with the effects (coefficients, SS, P-values) for the different factors.


DISCUSSION
L 330. I suggest that the following more cautious formulation is used: “…concentrations among the plant species tested, consistent with the hypothesis that glucosinolates stimulate feeding in C. cardariae.”
L 334. Include additional references supporting glucosinolates not showing strong phylogenetic signal, and also references that this pattern is not exclusive to glucosinolates but rather secondary defensive compounds (see comments on introduction).
L 343-344. Please provide details in methods as to how these normalizations were performed.
L. 372. “and if they are phylogenetically labile”

FIGURES AND LEGENDS
Table 2. The values for the ANOVA and KW tests should go in the legend, not as part of the table.
Fig. 1. It would be good to add a row for total GS per species.
Fig. 2. The symbol for mole is “mol”, so units should be expressed as “µmol/g”. It is best if significance is indicated in the graph panel.
Figs. 3 and 4. Make sure the figure legends are correct; seems to be switched, stating that figure 3 is glucosinolate dendrogram and figure 4 is genetic …
Table S1. The title for this table would be best to include “experimental design” in it, and highlight the five block used and referred to in the main text. This will help the reader focus on the main information of this table. Also, the date needs to include year.
Figs. S1. Are both young and old leaves cauline or are “old” leaves a mix of cauline and basal leaves?
+++

---

## Round 0.2 · Minor Revisions

Dear Dr. Rapo and colleagues:

Thanks for revising your manuscript. The sole reviewer is very satisfied with your revision (as am I). Great! However, there are a few minor concerns raised by the reviewer. Please address these ASAP so we may move towards acceptance of your work.

Best,

-joe

Reviewer 2 ·

Basic reporting

The authors have undertaken a careful and substantial revision, addressing all major concerns.

There are only small corrections should be done.

Minor comments
Line: 248 Throughout: ndhF should be italicized.
Line: 303 remove "with"
Line: 424 "disjunct"
Table2 column name "Total GS (nmol/g)*" might be µmol/g? In addition, what is the meaning of "*" shown here?

Experimental design

OK

Validity of the findings

OK

---

## Round 0.3 · accepted · Accept

Dear Dr. Rapo and colleagues:

Thanks for re-submitting your revised manuscript to PeerJ, and for addressing the concerns raised by the reviewer. I now believe that your manuscript is suitable for publication. Congratulations! I look forward to seeing this work in print, and I anticipate it being an important resource for research communities studying the ecology and evolution of herbivorous insects.

Thanks again for choosing PeerJ to publish such important work.

-joe